# Roles of Species-Specific Legumains in Pathogenicity of the Pinewood Nematode *Bursaphelenchus xylophilus*

**DOI:** 10.3390/ijms231810437

**Published:** 2022-09-09

**Authors:** Xi Zhang, Runmao Lin, Jian Ling, Yunsheng Wang, Feifei Qin, Junru Lu, Xin Sun, Manling Zou, Jing Qi, Bingyan Xie, Xinyue Cheng

**Affiliations:** 1College of Life Sciences, Beijing Normal University, Beijing 100875, China; 2Institute of Vegetables and Flowers, Chinese Academy of Agricultural Sciences, Beijing 100081, China; 3College of Plant Protection, Hunan Agricultural University, Changsha 410128, China; 4Ministry of Education Key Laboratory for Biodiversity Science and Ecological Engineering, Beijing 100080, China

**Keywords:** the pinewood nematode, plant parasitic nematode, comparative genome analysis, protease, legumain, functional complementation assay, *Arabidopsis* γVPE mutant, comparative transcriptome analysis

## Abstract

Peptidases are very important to parasites, which have central roles in parasite biology and pathogenesis. In this study, by comparative genome analysis, genome-wide peptidase diversities among plant-parasitic nematodes are estimated. We find that genes encoding cysteine peptidases in family C13 (legumain) are significantly abundant in pine wood nematodes *Bursaphelenchus* genomes, compared to those in other plant-parasitic nematodes. By phylogenetic analysis, a clade of *B. xylophilus*-specific legumain is identified. RT-qPCR detection shows that these genes are highly expressed at early stage during the nematode infection process. Utilizing transgene technology, cDNAs of three species-specific legumain were introduced into the *Arabidopsis* γvpe mutant. Functional complementation assay shows that these *B. xylophilus* legumains can fully complement the activity of *Arabidopsis* γVPE to mediate plant cell death triggered by the fungal toxin FB1. Secretory activities of these legumains are experimentally validated. By comparative transcriptome analysis, genes involved in plant cell death mediated by legumains are identified, which enrich in GO terms related to ubiquitin protein transferase activity in category molecular function, and response to stimuli in category biological process. Our results suggest that *B. xylophilu*-specific legumains have potential as effectors to be involved in nematode-plant interaction and can be related to host cell death.

## 1. Introduction

Peptidases have central roles in parasite biology and pathogenesis, and are very important and essential components of these processes. They degrade host proteins for nutrition, and manipulate the host immune system to elude the immune response [1]. Among them, cysteine peptidases in family C13 (clan CD), called legumains, have high specificity for the hydrolysis of acyl bonds of asparaginyl asparagine and play important roles in many physiological processes [2,3]. Legumains are widely distributed in animals and plants, with intriguing mechanistic peculiarities. In plants, they are called vacuolar processing enzymes (VPEs), they can help to process proteins in storage vacuoles, and are usually involved in plant cell death [4,5,6]. In the *Arabidopsis* genome, there are four VPEs (α, β, γ and δ VPEs), and the γVPE is specific to vegetative organs and associated with cell death processes [6,7]. In animals, legumain are also known as asparaginyl endopeptidases (AEPs) and have various functions including digestion, immunity, antimicrobial activity, immune signalling, apoptosis, transcription factor and osteoclast remodelling [8]. Legumains are also very important to parasitic protozoa and helminths [1]. It was reported that a surface-localized legumain appeared to display a pro-survival role in the intestinal protozoan *Blastocystis hominis*. Inhibition of legumain activity could induce programmed cell death (PCD) in *Blastocystis* [9]). In blood-feeding helminths, legumains were reported to be involved in host hemoglobin digestion [10,11]. Under some conditions, legumains are likely to be secreted to the pericellular environment. It was reported that a legumain was secreted to the blood of a host patient by the pathogenic parasite *Clonorchis sinensis* as an antigen to play an important role in host–parasite interaction [12]. However, little information is currently known about legumain in animal- and plant-parasitic nematodes.

The pinewood nematode (PWN), *Bursaphelenchus xylophilus*, is a migratory endoparasitic nematode, which lives in coniferous trees (mainly *Pinus* spp.) and causes a serious epidemic forest disease—the pine wilt disease (PWD). This invasive plant-parasitic nematode has killed millions of pine trees in its introduced regions in Asia and Europe, and has become a worldwide threat to forest ecosystem [13]. The nematode is transmitted to healthy trees by vector beetles (mainly pine sawyers, *Monochamus* spp.) during maturation feeding of the beetles [14,15]. Then, the nematode lives in the resin canals of host trees, and feeds epithelial cells and parenchyma cells [16,17]. An obvious characteristic of PWD is quick death of the living pine trees after infection with the nematode. Histopathological observation with electron microscopes (SEM and TEM) showed that, in inoculated pine seedlings, severe degradation in the resin canals, rays and the cambial zone had occurred before external symptoms appeared [17,18]. Cytological observations showed that, after inoculation of pathogenic PWN on Japanese black pine (*Pinus thunbergii*) seedings, the first conspicuous symptom was vacuolization of parenchyma cells, developed in a wide area and finally tonoplast burst. After the breakdown of vacuoles and the spread of vacuolar inclusion in the cells, parenchyma cells do not perform normal physiological function [19,20]. Moreover, the formation of vacuoles was observed before the nematode population increase in affected seedlings, indicating that the formation of vacuoles occurred without direct contact of nematodes with parenchyma cells. Therefore, it was suggested that some factors originating from the nematode *B. xylophilus* seemed to cause the formation and bursting of vacuoles in host [17,19]. For exploring the mechanisms of PWN pathogenesis, much work has been done and several hypotheses were proposed [15,21]. The genome of a Japanese strain of the nematode *B. xylophilus* has been published [22], and some parasitism-related genes and proteins involved in nematode-plant interactions have been reported [23,24,25,26]. However, the molecular mechanism by which such tiny worms of *B. xylophilus* kill such massive pine trees so rapidly are still unclear. 

In plant-pathogen interaction, plants have evolved immune systems that protect them from invading pathogens. Cell death is a defense mechanism of the host against biotrophic pathogens that occurs by inducing PCD, termed the hypersensitive response (HR), which may function to limit the spread of pathogens. Vacuole-mediated cell death is a plant defense strategy. VPE is a key molecule in the immune system that is associated with vacuolar membrane collapse [27]. However, pathogens can manipulate plant PCD pathways to their own advantage by inhibiting or inducing host PCD, which contributes to immunity, targeting defense compounds, or enhancing nutrient acquisition [28,29]. Cell death may instead be an infection strategy for some pathogens [30]. Plant pathogens have developed sophisticated attack strategies mediated by numerous effector proteins to stimulate cell death by hijacking the host cell’s defense machinery to promote the recycling of host cellular resources to absorb nutrients [31]. Plant VPE is also necessary for cell death mediation from a wide range of plant pathogens. Some necrotrophic pathogens can take advantage of the increased VPE activity in host cells to enhance pathogenicity [6,32]. Increased VPE activity may benefit the pathogen by mediating protein turnover and nutrient release [33]. The obligate biotrophic pathogen *Hyaloperonospora arabidopsidis* can also take advantage of the increased VPE activity in host cells to increase sporulation and enhance pathogenicity [6,32]. The hemibiotrophic pathogen *Fusarium moniliforme* produces mycotoxin FB1 to kill host cells through the host HR by the VPE-mediated vacuolar mechanism [5,34]. So far, the infection strategies by utilizing host cell death to their own advantage have been reported in fungal and bacterial pathogens. As the way of host cell death caused by *B. xylophilus* is similar to VPE-mediated PCD, it is an interesting question whether the nematode could use VPE activity to cause plant cell death.

With the increase in published genomes of plant-parasitic nematodes, it is possible to explore molecular host–parasite interactions by comparative genome analysis [35]. In this study, to explore the molecular pathogenic mechanism of PWN, we first try to explore diversities of peptidases in different plant-parasitic nematodes by comparing peptidases in plant-parasitic nematode genomes. Then, we try to identify *B. xylophilus*-specific legumain and estimate their potential for involvement in nematode-plant interaction by experimental verification. Our results may provide useful information regarding legumains in plant-parasitic nematodes.

## 2. Results

### 2.1. Diversity of Peptidases in Plant-Parasitic Nematodes by Comparative Genome Analysis

To explore the pathogenic mechanism of PWN, we firstly sequenced and assembled three *Bursaphelenchus* genomes of sibling species, i.e., *B. xylophilus* ‘R’ form (named BxCN), *B. xylophilus* ‘M’ form (named BxCA) and *B. mucronatus* (named BmCN), which represent strains with strong, weak, and non-virulence to pine hosts, respectively. The genome sizes of BxCN, BxCA, and BmCN are 79.2 Mb, 75.9 Mb, and 69.9 Mb, with 17,692, 17,941, and 16,661 predicted protein-coding gene models, respectively (Appendix A). The genome size and the gene number in BxCN are slightly different from the reported genome of the *B. xylophilus* ‘R’ form from the Japanese strain Ka4C1 (hereafter named BxJP), which is 74.56 MB in size with 18,074 protein-coding genes [22]. 

Then, utilizing available genome sequences deposited in NCBI (www.ncbi.nlm.nih.gov, accessed on 29 December 2019), 23 PPN genomes were download, and a total of 26 PPN genomes were used for comparison (Appendix A). These PPNs belong to 7 genera 17 species, which include 5 migratory endoparasites (7 genomes, 4 from *Bursaphelenchus*, 2 from *Ditylenchus* and one from *Radopholus similis*), 11 sedentary endoparasites (18 genomes, 13 from *Meloidogyne*, 3 from *Globodera* and 2 from *Heterodera glycines*), and one sedentary semi-endoparasite (*Rotylenchulus reniformis*). The genome sizes of PPNs vary greatly from ~38 Mb of *M. graminicola* [36] to ~313 Mb of *R. reniformis* [37,38], and the numbers of predicted protein-coding genes vary from 10,196 in the *M. graminicola* genome [36]) to 31,051 in the *M. enterolobii* genome [39] (Appendix A). Although a positive correlation between genome sizes and predicted gene numbers is found (R = 0.65, *p* < 0.01) (Appendix A), many predicted genes are incomplete as a result of low scaffold N50 values (<100 kb) in most PPN genomes.

Then, by BLAST searching predicted protein datasets of these PPN genomes against the complete MEROPS peptidase database (https://www.ebi.ac.uk/merops/), peptidases in each genome are identified (Appendix A). A total of 7861 genes are identified from the 26 PPN genomes, which encode peptidases assigned to 96 families of seven categories (A, C, M, N, P, S, T) according to their catalytic types. Among them, aspartic peptidases (A1), cysteine peptidases (C1, C19), metallo peptidases (M1, M13, M14, M41), serine peptidases (S16), and threonine peptidases (T1), are the abundant groups in the PPN genomes (Figure 1a, Appendix A). Comparison between migratory and sedentary PPNs shows that genes encoding peptidases in the migratory PPN genomes (371~601, mean 526.29) are much more abundant than those in the sedentary PPN genomes (42~330, mean 188.06), excepting those in the semi-endoparasitic nematode *R. reniformis* genome, which have 792 genes and 228 of them encode retropepsins (A2A) (Figure 1b, Appendix A). With software SignalP v4.0 [40] (https://services.healthtech.dtu.dk/service.php?SignalP-4.1), a total of 1625 secretory peptidases (20.67% of all peptidases) are predicted from the 26 PPN genomes (Appendix A). Notably, the percentage of secretory peptidases in the migratory PPNs (35.56% of total peptidases, mean 187.14 members in each) is significantly higher than those in the sedentary endoparasitic PPNs (6.59% of total, mean 12.39 members) and the semi-endoparasitic nematode *R. reniformis* (11.62%, 92 members) (Figure 1b, Appendix A). Among secretory peptidases, family A1 and C1 are the most abundant groups. In the four *Bursaphelenchus* genomes, a total of 578, 592, 538, and 525, peptidase-coding genes are identified from BxCN, BxJP, BxCA, and BmCN, respectively. Among them, genes encoding peptidases in family A1, C13, M1, M13, S10, and S33, are especially abundant in the *Bursaphelenchus* genomes, compared to other PPN genomes (Figure 1a, Appendix A). A total of 291 (50.35%), 250 (42.23%), 273 (50.74%), and 257 (48.95%), peptidases are predicted from BxCN, BxJP, BxCA, and BmCN, with signal peptides, respectively, which are putatively secretory peptidases. Notably, the percentages of secretory peptidases in four *Bursaphelenchus* genomes are significantly abundant in contrast to those in other PPNs (mean 267.75 vs. 25.18), especially peptidases in family A1, C13, M1, M13, S10 (Figure 1b, Appendix A). 

### 2.2. Expansion of Cysteine Peptidase in Family C13 in B. xylophilus and Their Evolutionary Relationship

We paid more attention to cysteine peptidases in family C13 (legumain/VPE/AEP), which has been documented to be involved in cell death in plants [6]. In the four *Bursaphelenchus* genomes, a total of 15, 16, 10, and 11, legumain-coding genes are identified from BxCN, BxJP, BxCA, and BmCN (Appendix A), and 13, 11, 10, and 11, of them are predicted to have signal peptides, respectively (Appendix A). However, only a total of 9 legumains are found in all other PPN genomes, which including one or two members in each of the genomes of *Ditylenchus* spp., *Globodera* spp., *R. similis,* and *R. reniformis*, but no homologue in the genomes of *Heterodera* spp. and *Meloidogyne* spp. (Appendix A). Additionally, three of them are predicted to have signal peptides (Appendix A). Obviously, legumain genes in the four *Bursaphelenchus* genomes are much more abundant than those in other PPN genomes, which are especially rich in the genomes of pathogenic *B. xylophilus* `R` form (BxCN and BxJP).

In addition to two homologues from the *Caenorhabditis elegans* genome, a phylogenetic tree of above PPN legumains (excluding five very short sequences) is constructed. The topology shows that these PPN peptidases are clustered to 12 orthologous groups (Figure 2a). Among them, two groups are conserved in PPNs. One group (BxCN04518 and orthologues) is homologous with *C. elegans* T05E11.6 (C13.005), belonging to glycosylphosphatidylinositol (GPI)-anchor transamidases. Another group (BxCN02210 and orthologues) is homologous with *C. elegans* T28H10.3 (C13.A02), belonging to hemoglobinase-type cysteine proteases. The other groups are composed of *Bursaphelenchus* members, with no homologue in other PPN genomes. A clade of *B. xylophilus*-specific legumains is identified (including BxCN10248 and BxCN10334 orthologous groups), which have no orthologue in *B. mucronatus* (Figure 2a). For further exploring the origin and evolutionary relationship of these legumains, we retrieved legumain sequences from all nematode genomes deposited in the NCBI database (including plant- and animal-parasitic nematodes, and free-living nematodes), in addition to plant VPEs from *A. thaliana*, *Pinus,* and *Picea* genomes. A phylogenetic tree of 89 members (with length of the C13 domain >200 aa) is constructed (Appendix A). The tree topology shows that the above two conserved groups (BxCN04518/C13.005 and BxCN02210/C13.A02 homologous groups) are also conserved in other nematodes. A large clade of *Bursaphelenchus*-specific legumains is found, with no homologue in other nematode genomes, and the *B. xylophilus*-specific legumains are involved in this clade (Appendix A). Based on the locations of legumain-coding genes in each of *Bursaphelenchus* genome, we find that some genes are tandemly arrayed (Appendix A). Sequence similarities and gene positions of legumains indicate gene duplication occurrence in the *B. xylophilus* genome (Figure 2b).

Comparison of primary structural organizations shows that *B. xylophilus*-specific legumain has a typic legumain structure, i.e., with a signal peptide, a peptidase_C13 domain (PF01650) and a C-terminal prodomain of legumain (legumain_C, cd2115), the same as that of human AEP and *Arabidopsis* γVPE (Figure 3a, Appendix A). Within the peptidase_C13 domain, four essential amino acid residues (R, H, C and S) exist, which form the substrate pocket, and the middle two residues (H and C) form the catalytic dyad [6]. The legumain_C domain is denoted as legumain stabilization and activity modulation domain (LSAM domain). The spatial structure of *B. xylophilus*-specific legumain (BxCN10334) predicted by RoseTTAFold software [41] ( https://robetta.bakerlab.org/) is also similar to that of human AEP and *Arabidopsis* γVPE (Figure 3b). Protein structure comparison by TM-align [42] (https://zhanggroup.org/TM-align/) shows that the TM-score value between *B. xylophilus*-specific legumain BxCN10334 and human AEP is larger than 0.87, and that between *B. xylophilus*-specific legumain and *Arabidopsis* γVPE is larger than 0.77 (Figure 3c). More structures and comparisons are shown in Appendix A. The results indicate high structural similarities among *B. xylophilus*-specific legumain, human AEP and *Arabidopsis* γVPE.

### 2.3. Expression Patterns of B. xylophilus-Specific Legumain Genes during the Nematode Infection in Pine Host

Through quantitative real-time PCR (RT-qPCR) technology, we detected the mRNA expression dynamics of the four species-specific legumains (BxCN10334, BxCN10337, BxCN10348, and BxCN10284) in *B. xylophilus* after the nematode inoculation on pine seedlings (*P. thunbergii*) for 0 (1–2 h) to 15 days. The result shows that these genes are highly expressed and reach a maximum after inoculation on the host for three to five days (Figure 4a). After inoculation for 10 days, their expressions decrease to very low levels. 

### 2.4. Functional Complementation of γVPE-Deficient Arabidopsis by Expression of B. xylophilus-Specific Legumains in Mediating Plant Cell Death

It has been documented that *Arabidopsis* γVPE can mediate mycotoxin FB1-induced cell death [5]. To explore the roles of *B. xylophilus*-specific legumains in host cell death, we first introduced two specific legumain genes (BxCN10334 and BxCN10337) into the *A. thaliana ∆γvpe* mutant (Salk_024036C) independently by genetic manipulation, with introduction of the empty vector pCAMBIA1302 as control. Then, leaves from five week old T3 plants were infiltrated with FB1 (10 mM, in 0.1% methanol) as an elicitor to trigger cell death. As expected, after FB1 infiltration for five days, necrotic lesions are clearly observed on treated leaves of the transgenic plants introduced with pCAMBIA1302::BxCN10334 and pCAMBIA1302::BxCN10337. These lesions are identical with the typical lesions showing cell death on leaves in the wild type *Arabidopsis* plants, while no obvious lesion is observed on leaves of the control plants introduced with the empty vector (Figure 4b).

Then, we quantitatively analyzed the effects of the four *B. xylophilus*-specific legumains (BxCN10334, BxCN10337, BxCN10284 and BxCA10290), in addition to a pine VPE (*Pinus taeda* PITA_000069534), on complement of *Arabidopsis* γVPE to mediate FB1-induced cell death, with the *∆γvpe* mutant and the wild type plants as negative and positive controls. A 10-μL of FB1 (10 mM, in 0.1% methanol) was infiltrated into each leaf of different *Arabidopsis* plants. After infiltration for five days, the diameters of necrotic spots on treated leaves of each transgenic line were measured (Figure 4c, Appendix A). The result shows that the mean sizes of lesions in transgenic lines with three BxCN genes (Φ 5.35–5.60 mm) are not only significantly larger than that in *Arabidopsis ∆γvpe* mutant plants (Φ 1.27 mm), but also larger than that in the wild type *Arabidopsis* plants (Φ 4.64 mm). The mean size in the transgenic lines of BxCA10290 (the orthologue in *B. xylophilus* ‘M’ form, Φ 2.81 mm) is obviously larger than that of the *∆γvpe* mutant, but smaller than that of the wild-type plants. The mean size of lesions in the transgenic lines with the pine *vpe* (Φ 4.67 mm) is close to that in WT *Arabidopsis* plants (Figure 4d). Statistical analysis shows that the differences among types of *Arabidopsis* plants are significant (df = 6, 278; F = 77.034, *p* < 0.001). A Tukey post-hoc test reveals significant pairwise differences between the *∆γvpe* mutant and each of the other types of *Arabidopsis* plants, as well as between BxCA10290 transgenic line and each of the other types plants (*p* < 0.001). The differences are not significant among three BxCN gene transgenic lines (BxCN10334, BxCN10337, BxCN10284), also between WT and the pine *vpe* transgenic line (Appendix A). The above results indicate that *B. xylophilus*-specific legumains in the pathogenic ‘R’ form (BxCN) can fully recover the activity of the *Arabidopsis ∆γvpe* mutant in mediating cell death, but the role of the orthologue in the ‘M’ form nematode (BxCA) is relatively weak.

### 2.5. Validation of Secretory Activities of B. xylophilus-Specific Legumains

To determine whether these *B. xylophilus*-specific peptidases can be secreted into host cells by stylet to play roles, different experiments were performed to validate their secretory activities, which are necessary for effectors to interact with host. We first analyzed the tissue localization of the mRNA expression of three *B. xylophilus*-specific legumains (BxCN10334, BxCN10337, and BxCN10284) in the nematode, by in situ hybridization. With digoxigenin-labelled antisense cDNA probes, strong signals are observed within the oesophageal glands in juvenile and adult nematodes (Figure 5a). Similarly, using fluorescence in situ hybridization, strong fluorescent signals are detected in the subventral and dorsal oesophageal glands in juveniles and adults, respectively (Figure 5b). No signal is detected using the sense cDNA probes.

Then, secretory activities of their signal peptides are validated by using a genetic assay based on invertase secretion for yeast growth on plates with sucrose or raffinose as the sole carbon source. Sequences of predicted signal peptides of the above three genes (BxCN10334, BxCN10337, and BxCN10284) were inserted into the plasmid pSUC2T7M13ORI, and then transferred into the yeast strain YTK12, respectively. The result shows that all transformants can grow on YPRAA media and display red color with the chemical 2,3,5-triphenyltetrazolium chloride (TTC) (Figure 6). 

### 2.6. Identification of Genes Involved in Nematode Legumain-Mediated FB1-Induced Plant Cell Death

To determine which genes are involved in nematode legumain-mediated FB1-induced plant cell death, we sequenced and compared transcriptomes of four types of *Arabidopsis* plants, i.e., *∆γvpe* mutant, the wild type, transgenic lines introduced with BxCN10334, and BxCN10337, each with two replicates. By comparison with gene expression in the mutant transcriptomes, a total of 910, 2442, and 1958, differentially expressed genes (DEGs) are identified from the transcriptomes of the wild type, the transgenic lines introduced with BxCN10334 and BxCN10337 *Arabidopsis* plants, respectively (*p* < 0.05 and |log2FC| ≥ 1). Among them, 59 DEGs display similar expression patterns (26 up-regulated and 33 down-regulated) in the transcriptomes of wild type and the two transgenic lines, contrast to the transcriptome of *∆γvpe* mutant (Figure 7a, Appendix A). Among 26 upregulated DEGs, four encoding E3 ubiquitin ligases (BRH1, ILP, BRG1, BRG2) and two encoding F-box proteins (EBF2, AFB2) are putatively involved in protein ubiquitination and proteasome-mediated ubiquitin-dependent protein catabolic process; three encoding protein kinases (PID, PK1B, PPCK1) and one encoding an ethylene response factor (ERF106) are possibly involved in signal transduction; one (AT2G30230) encoding a 6,7-dimethyl-8-ribityllumazine synthase is involved in response to fungus stimulus, and three encoding proteins (DIN10, AT2G44670, AT5G47060) are putatively involved in response to stress, such as chemical, starvation, ethylene stimulus, and so on; two encoding transcription factors (NAC041, AT3G50650) are involved in regulation of transcription; one ADT3 encoding arogenate dehydratase may be involved in amino acid metabolism; one BGLU16 encoding beta-glucosidase 16 may be involved in carbohydrate metabolism; others encode proteins including a kinase regulator (MAKR6), a F-box protein (PP2-A12), three transporters (MFS1, AT1G16500, AT1G79160), and three functional unknown proteins (AT2G27770, AT3G55646, AT5G62280). Among those 33 downregulated DEGs, they encode proteins including five transcription factors (ERF109, ERF022, CBF4, MYB75, WRKY33), two ARM repeat superfamily proteins (AT3G02840, AT5G58680) and one kelch repeat superfamily protein (AT4G39570), two kinases (CRK10, ZRK15), two F-box proteins (AT3G03030, AT2G27310), two U-box proteins (PUB5, PUB19), one E3 ubiquitin ligase (ATL80), one cytochrome P450 (CYP707A3), one hydrolase (NUDT4), and 16 other proteins (Appendix A). GO enrichment analysis by PANTHER (with *p*-value cutoff of 0.05) shows that above 59 DEGs are enriched in GO terms related to ubiquitin-(like) protein transferase activity (GO:0004842, GO:0019787) in category molecular function (MF), and in terms related to response to endogenous stimuli (GO:0009719) or chemical stimuli (GO:0042221) in category biological process (BP), including response to endogenous hormone stimulus (GO:0009725, GO:0032870) and hormone-mediated signaling pathway (GO:0009755), as well as response to chemical organic substance (GO:0010033) and oxygen-containing compound (GO:1901700) (Figure 7b, Appendix A). 

Moreover, we also identify an additional 1333 DEGs displaying similar expression patterns (up- or down regulated) in transcriptomes of the two transgenic lines introduced with BxCN10334 and BxCN10337, including 261 significantly upregulated and 1072 downregulated genes (Figure 7a, Appendix A). GO enrichment analysis shows that these DEGs are enriched in GO terms response to stimulus (GO:0050896), such as to chemical (GO:0042221), stress (GO:0006950), and hormone (GO:0009725), and terms biological regulation (GO:0065007) and signal transduction (0007165) in category BP (Figure 7c, Appendix A). 

We then randomly selected 39 DEGs (including 16 upregulated and 23 downregulated) from the 59 common DEGs to verify their expression patterns in the four types of *Arabidopsis* plants (i.e., *∆γvpe* mutant, wild-type, and transgenic plants introduced with BxCN10334 and BxCN10337) by RT-qPCR. The results show similar expression patterns as those observed in the transcriptome profiles (Appendix A).

## 3. Discussion

Although legumains are documented to play important roles in animals and plants, little is known in nematodes, currently. In this study, by comparative genome analysis, we find that genes encoding legumains are significantly abundant in the pathogenic *B. xylophilus* ‘R’ form genome, in contrast to those in other plant-parasitic nematode genomes (Figure 1 and Figure 2, Appendix A). Based on sequence similarities and the positions of legumain genes in the *B. xylophilus* genome, we suggest that the expansion of legumain-encoding genes in the *B. xylophilus* genome is mainly contributed by gene duplication (Figure 2b). We identified four *B. xylophilus*-specific legumains, which have similar structures to *Arabidopsis* γVPE and human AEP (Figure 3, Appendix A). In general, proteins with high sequence identity and high structural similarity tend to possess functional conserved [43,44]. As both human AEP and *Arabidopsis* γVPE are related to cell death [6], *B. xylophilus*-specific legumains with high sequence identities and high structural similarities to them perhaps also have similar function in cell death. Then, utilizing genetic manipulation, genes encoding *B. xylophilus*-specific legumains were transferred to *A. thaliana ∆γvpe* mutant, and a functional complementation assay was performed. After infiltration of fungal toxin FB1, necrotic lesions are clearly observed on treated leaves of the transgenic plants introduced with *B. xylophilus*-specific legumain genes, which are similar to the lesions on leaves in the wild type *Arabidopsis* plants (Figure 4b,c, Appendix A). The result indicates that these *B. xylophilus*-specific legumains can fully recover the activity of *Arabidopsis* γvpe mutant to mediate plant cell death triggered by the fungal toxin FB1. Of cause, further verification is necessary to obtain supports in the pine host tree. However, owing to the limitation of current technology, it is difficult to perform experimental verification of nematode legumains in a pine tree by host-mediated RNAi or by gene knockout in the nematode. An interesting question in this study is that the mean size in the transgenic lines of BxCA10290 (the orthologue in *B. xylophilus* ‘M’ form) is obviously smaller than that of BxCN10334 (the orthologue in *B. xylophilus* ‘R’ form). We compared the protein structures of BxCN10334 and BxCA10290, and their identity of amino-acids is 93% and TM-score > 0.92 (Appendix A). Limited by our current knowledge, we cannot interpret the virulence difference from protein structures; more study is needed in the future. Using recombinant DNA technology, we may explore the biochemical characteristics of these legumains through heterologous expression of these legumain genes in a bacterium or yeast.

Moreover, hybridization in situ verify that these *B. xylophilus*-specific legumains are expressed in esophageal glands of the nematode (Figure 5). Yeast invertase secretion assay validates the function of the signal peptides harbored in the legumains (Figure 6). Gene expression by RT-qPCR detection shows that these *B. xylophilus*-specific legumain genes are expressed at the early stage of the nematode infection (Figure 4a). The results indicate that these nematode legumains have effector potentials. Based on the above results, we suggest that these species-specific legumains are secretory proteins, which are likely to be secreted out through the nematode stylet into the host cells, and may play roles as effectors to participate in pathogen-plant interactions in the early stage during nematode infection. In consideration of previously histopathological and cytological results [17,18,19], a molecular mechanism of *B. xylophilus* killing pine trees is proposed. It is that, when the nematode enters a healthy pine tree and feeds on the host’s parenchyma and epithelial cells, the host immune system may be activated, and a defensive HR mediated by VPE be produced. Meanwhile, the worms may secrete species-specific legumains through their stylets when they are feeding, which have plant VPE activities and may strengthen plant cell death, resulting in a pathogenic PCD development in susceptible pine trees. However, no pathogenic PCD occurs in resistant pine trees. Perhaps it is owing to that the hosts can slow damage expansion in tissues early and have time to complete the biosynthesis of lignin in the cell walls, as reported previously [45]. Lignin surrounds the damaged regions and induce cell wall protein-based defense. Based on our current results, we suggest that the pathogenic nematode *B. xylophilus* may uses a strategy like some fungal phytopathogens, by inducing plant PCD to cause inappropriate cell death in hosts for improving their parasitism [46]. With biotechnological developments, nematode genes may be introduced into a pine host by genetic manipulation, and host-mediated RNAi may be performed. Alternatively, *B. xylophilus* mutants may be obtained by gene knock-out technology. Then, the biological functions and mechanisms of different legumains in the nematode will be determined.

## 4. Experimental Procedures

### 4.1. Nematode Strains

Three *Bursaphelenchus* nematode strains were used for whole genome sequencing in this project, i.e., *B. xylophilus* ‘R’ form strain ZJSS and *B. mucronatus* strain BmDH from *P. massoniana* in Zhejiang Province, China, and *B. xylophilus* ‘M’ form strain CAN from Canada (a gift from Dr. HM. Li). The three nematodes are similar in morphology, biology, and ecology. Therefore, they were once taken as a “super species” or the pinewood nematode species complex (PWNSC) [47,48]. However, the pathogenicity to pine hosts is obviously different [49,50]. Nematodes were cultured on fungal mats of *Botrytis cinerea* grown on potato-dextrose agar (PDA) plates at 25 °C. After sister-brother mating over 10 generations for each strain, established inbred lines were prepared for genome sequencing. 

### 4.2. Plant Materials

Seeds of *Arabidopsis thaliana* Col-0 ecotype and the *∆γvpe* mutant Salk_024036C (from the Nottingham Arabidopsis Stock Centre, NASC) were surface-sterilized with 75% ethanol, and then sown onto 0.4% agar containing 1/2 Murashige-Skoog (MS) medium and 1% sucrose for three days in the dark at 4 °C for vernalization as described [5]. Seeds were germinated and grown in a growth chamber at 22 °C under a 16 h light/8 h dark period for two weeks. Then, seedlings were transplanted in cultivated matrix and grown under the same conditions. These *Arabidopsis* plants were used for genetic transformation and functional complementation assays. *Nicotiana benthamiana* were cultured in a greenhouse and used for transient expression. Five year old pine seedlings (*P. thunbergii*) were prepared for nematode inoculation. 

### 4.3. Genome Sequencing and Assembly

Genomic DNA was extracted from each of the inbred lines of above three nematodes at mixed stages using a regular phenol chloroform method. For effective genome assembly, multiple libraries with different insert sizes were constructed for each nematode strain (Appendix A). Genome sequencing was performed using an Illumina GA II sequencing system with standard procedures at the Beijing Genomics Institute (Shenzhen, China). In total, 9.7 Gb, 14.4 Gb, and 12.1 Gb, were generated for *B. xylophilus* ‘R’ form (named BxCN), *B. xylophilus* `M` form (named BxCA), and *B. mucronatus* (named BmCN), respectively (Appendix A).

After filtered to remove low-quality sequences, base-calling duplicates and adaptor contamination, the sequence data from each library were assembled using SOAPdenovo with default parameters [51]. We first assembled the reads from the short insert-size libraries (≤500 bp) into contigs using *k*-mer overlap information (*de Bruijn* graph *k*-mer). A hierarchical assembly method was used to construct scaffolds step by step by adding data from each library separately in the order of insert size from smallest to largest. Gaps were filled in by following the methods described previously [52]. Then, taking assembly of the published genome of *B. xyolphilus* strain Ka4C1 as the reference [22], genome assemblies of the two *B. xylophilus* genomes (BxCN and BxCA) were improved by using mugsy [53]. We assessed the completeness of the draft genomes using the Core Eukaryotic Genes Mapping Approach (CEGMA) [54], and annotated repetitive elements in each genome by RepeatScout [55] and RECON [56].

### 4.4. Protein-Coding Gene Annotation

The protein-coding genes were inferred using de novo-, homology- and evidence-based approaches. De novo gene prediction was performed on repeat-masked genomes using four gene prediction programs (Augustus, GlimerHMM, SNAP, and GeneMark.hmm-ES) [52]. Training gene model sets were generated from subsets of the EST/mRNA and RNA-Seq datasets representing 822 distinct genes of BxCN and 808 distinct genes of BmCN. The same dataset used for BxCN was also used for BxCA. Taking advantage of the published protein-coding gene annotation of Ka4C1 [22], we used the homology-based gene annotation program genBlastG [57] to find potentially missed gene models in BxCN, BxCA, and BmCN. RNA-seq reads obtained by sequencing the reverse transcribed mRNA libraries of BxCN, BxCA, and BmCN, were aligned using Bowtie [58]. Introns were defined using TopHat [59].

Following the prediction of the protein-coding gene set, each inferred amino acid sequence was assessed for conserved protein domains in Pfam v31.0 [60] and NCBI CDD database [61]. SignalP v4.0 [40], TargetP v1.1 [62] were applied to predict signal peptides, and SignalP, Phobious, and TMHMM v2.0c [63] were applied to predict transmembrane domains in the proteins. We predicted peptidases by aligning amino acids against the reported peptidases in the MEROPS database [3] with an E-value cutoff of 1 × 10^−5^. The predicted peptidases were further confirmed and used for following analysis if they contained conserved domains. For the peptidases from PPN genomes, we used TBtools [64] (Chen et al., 2020) to draw the heatmap. The 3D structure of legumain is predicted by RoseTTAFold server [41] (https://robetta.bakerlab.org). Protein structure comparison is performed by TM-align [42], i.e., uploading the PDB files of protein structure to the TM-align server (https://zhanggroup.org/TM-align/) for comparative analysis. The TM-score of the comparative analysis will be provided by the TM-align method and the PDB files for pair-wise structure comparison will be generated by TM-align. To view the protein structure, we used iCn3D [65] and Jmol v14.32.70 (http://jmol.sourceforge.net/).

We also downloaded all available PPNs genomes (23 genomes) deposited in the public NCBI databases, in addition to *C. elegans* genome (Appendix A). As some genomes had no gene information, we performed de novo prediction to identify the conserved genes using Augustus [66] (https://bioinf.uni-greifswald.de/augustus/) with the training dataset of *C. elegans*.

### 4.5. Phylogenetic Tree Construction and Evolutionary Analysis

For the phylogenetic analysis of PPN legumains, we used both the maximum likelihood (ML) method with WAG + G model and the neighbor-joining (NJ) method with JTT model. Alignment of sequences was performed using the program MUSCLE [67]. The neighbor-joining phylogenetic tree was constructed based on whole gene sequences. Prottest [68] was used to evaluate the best models for the maximum likelihood phylogenetic tree, which was constructed based on the C13 domain sequences with length larger than 110 aa. The MEGA v6 [69] was used to construct the gene trees. Then, we explored homologous genes of all nematode legumains on NCBI, in addition to plant VPEs (*A. thaliana* and conifers), and a phylogenetic tree of 89 members (length of the C13 domain >200 aa) was constructed by the (ML) method with the LG + I + G model. 

### 4.6. RNA Isolation, RT-PCR and Quantitative Real-Time PCR

Fresh cultured nematodes were collected and the total RNA was isolated using TRIzol reagent (Invitrogen, Carlsbad, CA, USA). After measuring the concentration (OD260) and the purity (OD260/OD280 1.8~2.0) with an ultraviolet spectrophotometer, agarose gel electrophoresis was carried out to check the RNA integrity. Then, 1 μg of RNA was used for first-strand cDNA synthesis using the SuperScript™ III cDNA Synthesis Kit (Invitrogen, Carlsbad, CA, USA), according to the manufacturer’s instructions. Using cDNA as the template, RT-PCR was performed with primer-pairs (Appendix A). PCR conditions were as follows: 98 °C for 1 min, 35 cycles of 98 °C for 10 s, Tm °C 20 s, and 72 °C for X s (set up as 60 s/kb), and finally, 72 °C for 5 min. Then, PCR products were purified with a gel purification kit (Tiangen, Beijing, China) and Sanger-sequencing by forward and reverse primers. Gene sequences of predicted *B. xylophilus*-specific legumains were validated. 

For detection of the mRNA expression dynamics during nematode infection, we inoculated the nematode *B. xylophilus* on five year old pine seedlings (*P. thunbergii*), with approximately 10,000 individuals (200 μL) for each. After inoculation for 0 (1–2 h), 1, 3, 5, 10, and 15 days, nematodes were isolated, respectively. Total RNA was extracted and first-strand cDNA was synthesized as above for each treatment, independently, with at least three replicates for each treatment. RT-qPCR was performed with primers (Appendix A) to detect expression levels of *B. xylophilus*-specific legumain-coding genes. Each 20 μL reaction mixture was prepared using a SYBR Premix ExTaqTM II kit (Takara, Dalian, China) on a CFX96 Real-Time PCR System (BIO-RAD, CA, USA) following the manufacturer’s protocol. Cycling conditions for quantitative PCR were: 95 °C for 30 s, and 45 cycles of 95 °C for 5 s, 60 °C for 20 s and 72 °C for 15 s. Each treatment had three independent repeats, with four technical replicates for each reaction. The ef1α gene was used as the internal control. The relative gene expression changes were calculated using the 2^−ΔΔCT^ method.

### 4.7. Functional Complementation Assay

Nematode legumain-coding genes (with removed signal peptides) were amplified from the cDNA templates with primers (Appendix A), then inserted into the plant transformation vector pCAMBIA1302. The resulting constructs were independently introduced into the *Arabidopsis ∆γvpe* mutant (Salk_024036C) through *Agrobacterium*-mediated transformation following a method described previously [70]. A transformant with the empty vector pCAMBIA1302 was used as a control. All transformants were then incubated overnight in the dark at 25 °C and moved to a greenhouse under controlled conditions (14 h light/10 h dark at 25 °C) for growth and seed harvesting. Seeds of each generation were screened on MS medium plates with hygromycin B (100 mg/mL). All putative transformants were confirmed by PCR amplification with the primer pair (pCAMBIA1302-F/-R, Appendix A) from genomic DNA isolated from the leaves using a DNeasy Plant Mini Kit (QIAGEN, Hilden, Germany) following the manufacturer’s instructions. Homozygous T3 plants were used for further experiments.

We first introduced two legumain-coding genes (BxCN10334 and BxCN10337) into the *Arabidopsis ∆γvpe* mutant for the complementation assay. Homozygous transgenic seedlings expressing 35S::BxCN10334 or 35S::BxCN10337 were grown for five weeks. Then, detached leaves were infiltrated with 10 mM FB1 (in 0.1% methanol) as described previously [5]. Lesion development on leaves was observed after five days. Then, four *B. xylophilus*-specific legumain-coding genes (including three BxCN and one BxCA) and a plant *vpe* gene from *P. taeda* (*PITA_000069534*) were introduced into the mutant independently. For quantitative comparison of complemental effects, a 10-μL of FB1 (in 0.1% methanol) was infiltrated into each leaf. After five days, the diameters of necrotic spots on the treated leaves were measured under a light microscope (Olympus, Tokyo, Japan). More than 10 plants and each with 3–4 leaves were treated with FB1 infiltration in each transgenic line. The mean size of each line was calculated. One-way ANOVA was used to compared the mean sizes with the SPSS software v20 (IBM, Chicago, IL, USA).

### 4.8. In Situ Hybridization 

We used both digoxigenin-labelled probes method [71] and the fluorescein isothiocyanate method [72] to determine tissue expression of the three *B. xylophilus*-specific legumains (BxCN10334, BxCN10337, and BxCN10284) in the nematode. For digoxigenin-labelled probe hybridization, a fragment of approximately 300 bp was amplified from the coding regions of each gene and used as a template for the synthesis of both DIG-labelled sense and antisense probes (Appendix A). Hybridization was performed with mixed life stage nematodes. Hybridization signals within the nematodes were detected with alkaline phosphatase-conjugated anti-digoxigenin antibody and substrate and observed under a light microscope (Olympus, Tokyo, Japan). For fluorescence probe hybridization, an approximately 25 bp cDNA fragment specific to each gene was used as an antisense and sense probe, and the 5′- end was labelled with fluorescein isothiocyanate (FITC, Sangon Biotech, Shanghai, China) as previously described with modification [72]. The hybridization signals were observed under a confocal microscope (Zeiss LSM700, Oberkochen, Germany). Probe sequences are listed in Appendix A.

### 4.9. Secretory Activity Assay 

The yeast secretion system was used to validate the function of signal peptides [73]. The signal peptide sequences of above three legumains (BxCN10334, BxCN10337, and BxCN10284) were independently inserted into the yeast signal sequence trap vector pSUC2T7M13ORI. Then, the constructs were transformed into the invertase-negative yeast strain YTK12 using the Frozen-EZ Yeast Transformation II Kit (Zymo Research, CA, USA). After transformation, the yeast was plated on CMD-W plates for three days. Yeast colonies were purified and then grown on raffinose-containing YPRAA plates. The YPDA plates were used as equal viability controls. In addition, invertase enzymatic activity was detected with 2,3,5-triphenyltetrazolium chloride (TTC) as described in a previous study [74].

### 4.10. RNA Sequencing and DEG Identification 

Using TRIzol reagent (Invitrogen, Carlsbad, CA, USA), following the manufacturer’s instructions, total RNA was extracted independently from leaves of four types of *Arabidopsis* plants, i.e., ∆γvpe mutant, wild-type, and transgenic lines with BxCN10334 and BxCN10337, which were infiltrated with FB1 for five days to induce cell death. At least 20 mg of total RNA was prepared from each sample. Library construction and RNASeq were performed at Wuhan Frasergen Bioinformatics Co. (Wuhan, China), using Illumina HiSeq 2000. Two independent replicates were prepared for sequencing. After removing adaptor sequences and low-quality reads, the clean reads (36371964~54419250 reads, 5455~8162 Mb) were mapped to the *Arabidopsis* genome (TAIR10, GCA_000001735.1), with TopHat (-r 30 -p 20). To obtain gene expression FPKM values, Cufflinks and TopHat were used for analysis as shown in the reported workflow [75]. DEGs were obtained using the DEseq package [76] by comparing gene expression profiles of transgenic lines and WT with that of *∆γvpe* mutant (using the mean value of two replicates, |log_2_Ratio| > 1, *p*-value < 0.05, FDR < 0.01). Shared DEGs with identical patterns (significantly upregulated or downregulated in WT and two transgenic lines, compared to the mutant control) were selected for GO enrichment analysis. Enrichment analysis of DEGs was performed by PANTHER [77], with a *p*-value cutoff of 0.05 and two interactive tools (overrepresentation test and enrichment test). 

For verification of the expression patterns, 39 DEGs (16 up- and 23 downregulated) were randomly selected from the shared DEGs, and RT-qPCR was performed using the four *Arabidopsis* cDNA templates (i.e., ∆γvpe mutant, wild-type, and transgenic plants with BxCN10334 and BxCN10337) as above. At least three replicates for each gene and four technical replicates for each reaction were performed. The *actin2* (AT3G18780) gene was used as the internal control gene. Primers are listed in Appendix A.

## Figures and Tables

**Figure 1 ijms-23-10437-f001:**
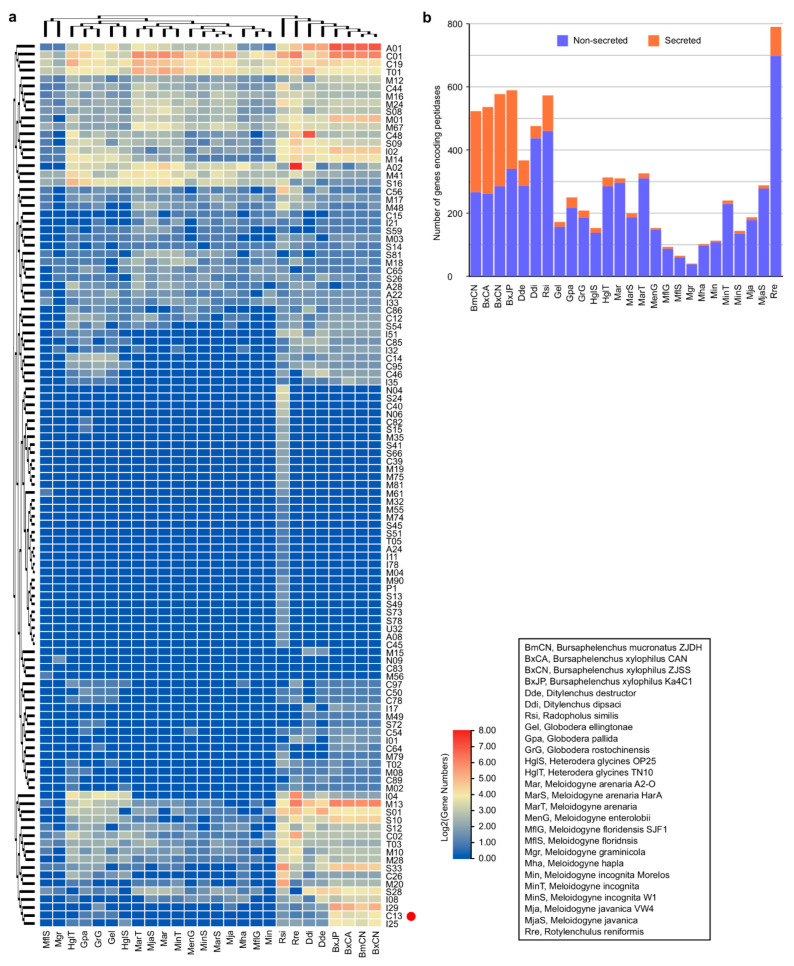
Predicted genes encoding peptidases in plant-parasitic nematode genomes: (**a**) heatmap of peptidases distribution. The red dot marks the peptidase C13 family (legumains); (**b**) number of peptidases in each genome.

**Figure 2 ijms-23-10437-f002:**
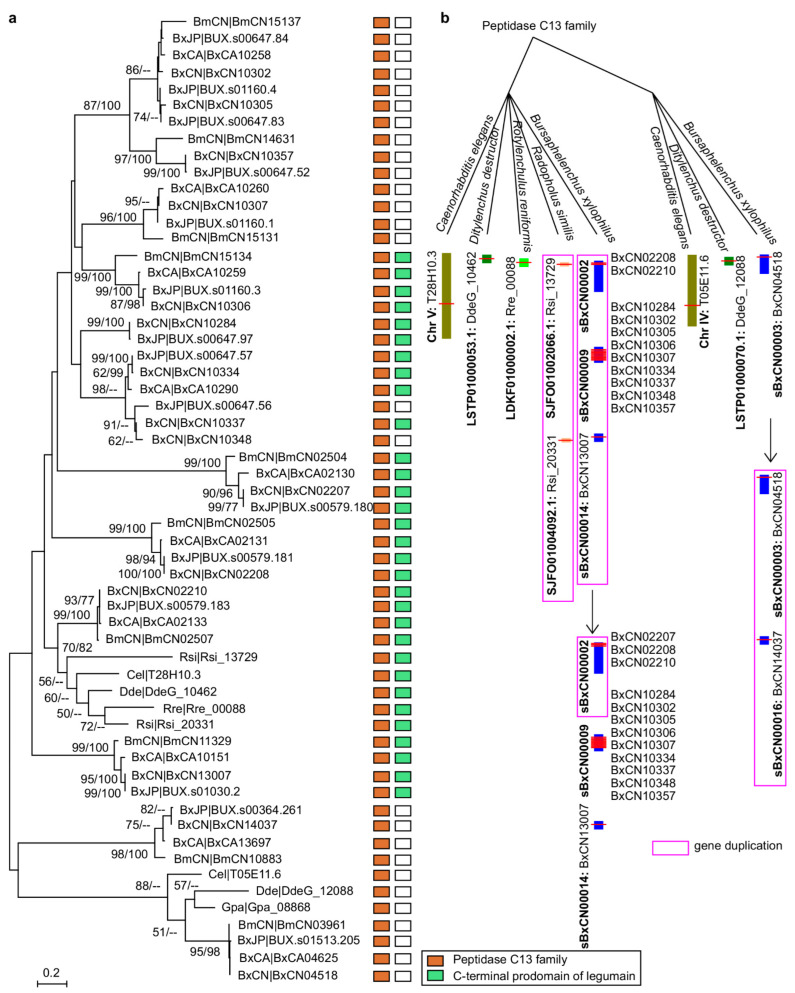
Phylogenetic relationship of legumains in plant-parasitic nematodes and *C. elegans*: (**a**) phylogeny of nematode legumains and their primary structures, i.e., a peptidase_C13 domain (PF01650); and (**a**) C-terminal prodomain of legumain (legumain_C, cd2115). The phylogenetic tree was constructed based on full-length protein sequences. Bootstrap values of ML/NJ are showed on branches, and bootstrap value less than 50% is replaced by the marker --. The *B. xylophilus*-specific legumains are enclosed in the red box; (**b**) putative duplications of legumain-coding genes in the *B. xylophilus* genome (BxCN).

**Figure 3 ijms-23-10437-f003:**
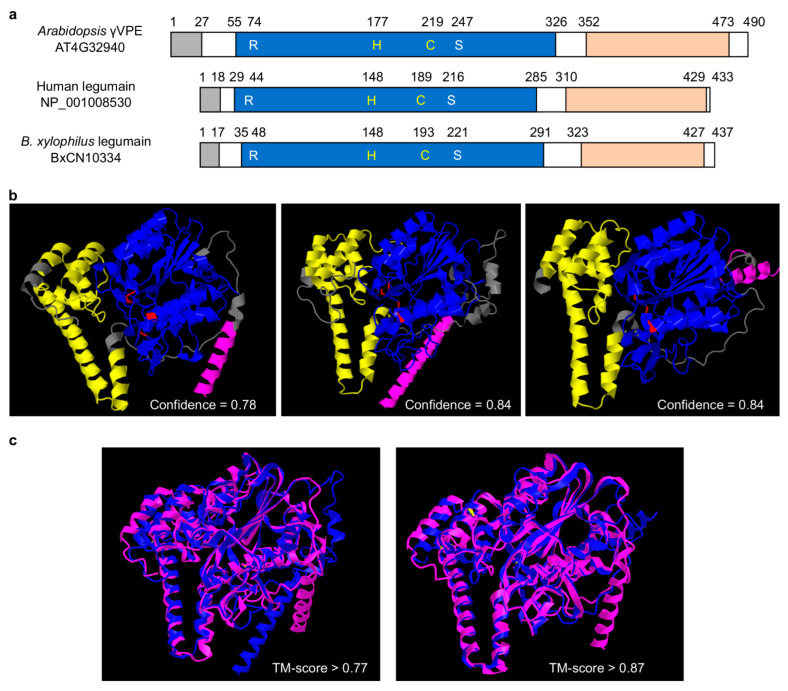
Structural comparison of legumains: (**a**) primary protein structural organization of *B. xylophilus*-specific legumain (BxCN10334), *Arabidopsis* γVPE (AT4G32940), and human AEP (NP_001008530). They have a signal peptide (gray box) at the N-terminus, a peptidase_C13 domain (PF01650) (blue box) and a C-terminal prodomain of legumain (legumain_C, cd2115) (orange box). The C13 domain includes four essential amino acid residues (R, H, C, S), which form the substrate pocket, and the middle two (H, C) are catalytic dyad; (**b**) prediction of spatial structures of *B. xylophilus*-specific legumain, *Arabidopsis* γVPE, and human AEP, with RoseTTAFold online (job IDs 207107, 207108 and 207109, respectively). Pink: signal peptide; blue: peptidase C13 family domain; yellow: C-terminal of prodomain of legumain; red: the essential amino acid residues; (**c**) protein structure comparison by TM-align, between *B. xylophilus* legumain and *Arabidopsis* γVPE, *B. xylophilus* legumain and human AEP. Pink represent *B. xylophilus* legumain, blue represent *Arabidopsis* γVPE or human AEP.

**Figure 4 ijms-23-10437-f004:**
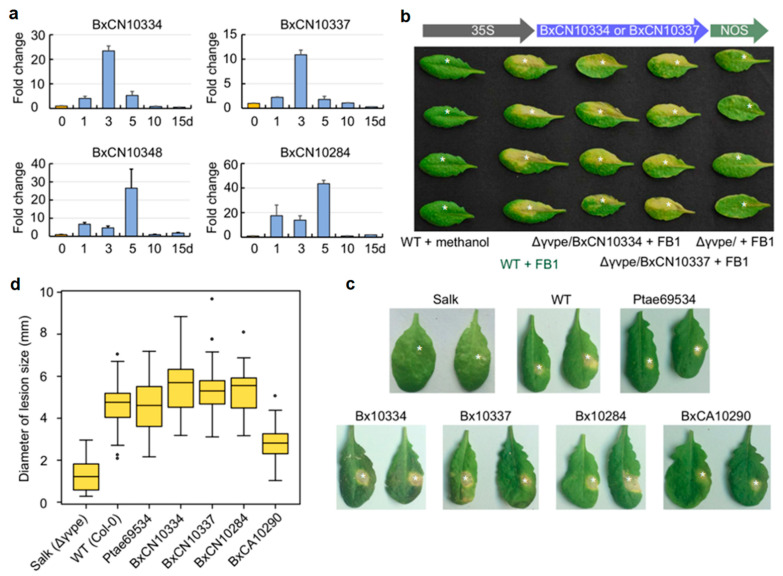
Expression patterns and functional complementation assay of *B. xylophilus*–specific legumains: (**a**) RT-qPCR detection of mRNA expression of *B. xylophilus*-specific legumains after the nematode was inoculated on five years old pine seedlings (*P. thunbergii*) for 1 to 15 days, taking 0 d (1–2 h) as control. The EF-1α gene was used as an internal control. Bars are standard errors (SE); (**b**) two BxCN legumain genes (BxCN10334, BxCN10337) were introduced into the *Arabidopsis Δγvpe* mutant (Salk_024036C). Necrotic spots are observed on leaves of homozygous T3 plants after infiltration with FB1 (10 mM, in 0.1% methanol) for five days, which are similar to those in the wild type *Arabidopsis* plants infiltration with FB1. The mutant plants infiltrated with FB1 and the wild type plants infiltrated with methanol (solvent) are taken as controls; (**c**) quantitative assay of complementation effects of transgenic lines with four *B. xylophilus*-specific legumains and a pine VPE (*Pinus taeda* PITA_000069534), by infiltration with 10 μL of FB1 (in 0.1% methanol) on each leaf. Necrotic lesions on *Δγvpe* mutant and the wild type *Arabidopsis* plants are taken as controls; (**d**) comparison of the mean diameters of necrotic spots in different transgenic lines and the wild-type *Arabidopsis* plants, compared with the *Δγvpe* mutant. Black dots represent outliers.

**Figure 5 ijms-23-10437-f005:**
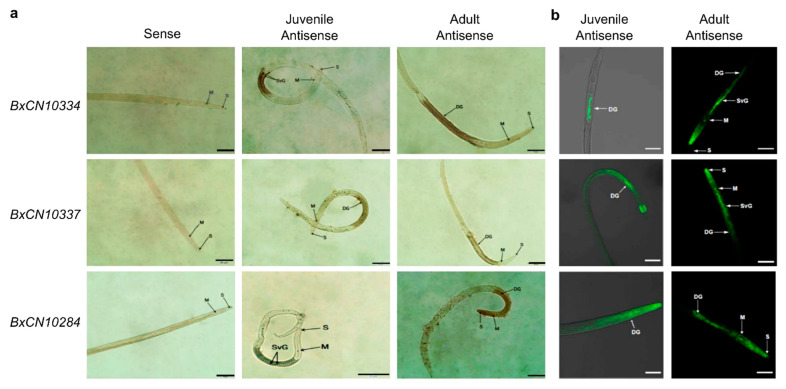
Localization of three BxCN legumain transcripts in the nematode by in situ hybridization: (**a**) hybridization with digoxigenin-labelled cDNA probes. Scale bar, 50 μm; (**b**) hybridization with 5′- end FITC-labelled cDNA probes. DG, dorsal glands; M, median bulb; S, stylet; SVG, subventral glands. Scale bar, 20 μm.

**Figure 6 ijms-23-10437-f006:**
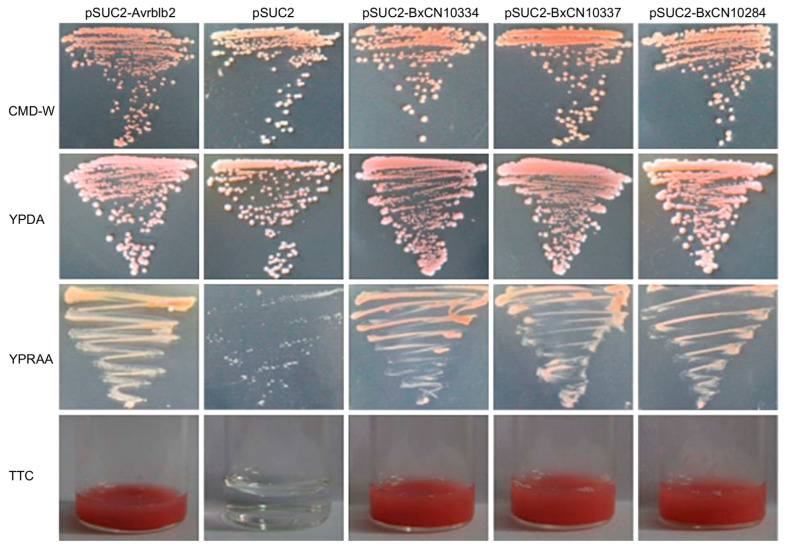
Yeast invertase secretion assay of the signal peptides of the above three BxCN legumains. Yeast YTK12 strain carrying signal peptides of BxCN10334, BxCN10337, and BxCN10284, are able to grow in raffinose-containing YPRAA medium, and react with TTC to display red color. YTK12 carrying the Avrblb2 and the pSUC2 vector are used as positive and negative controls, respectively. CMD-W, YPDA, and YPRAA, are culture media. TTC, 2,3,5-triphenyltetrazolium chloride.

**Figure 7 ijms-23-10437-f007:**
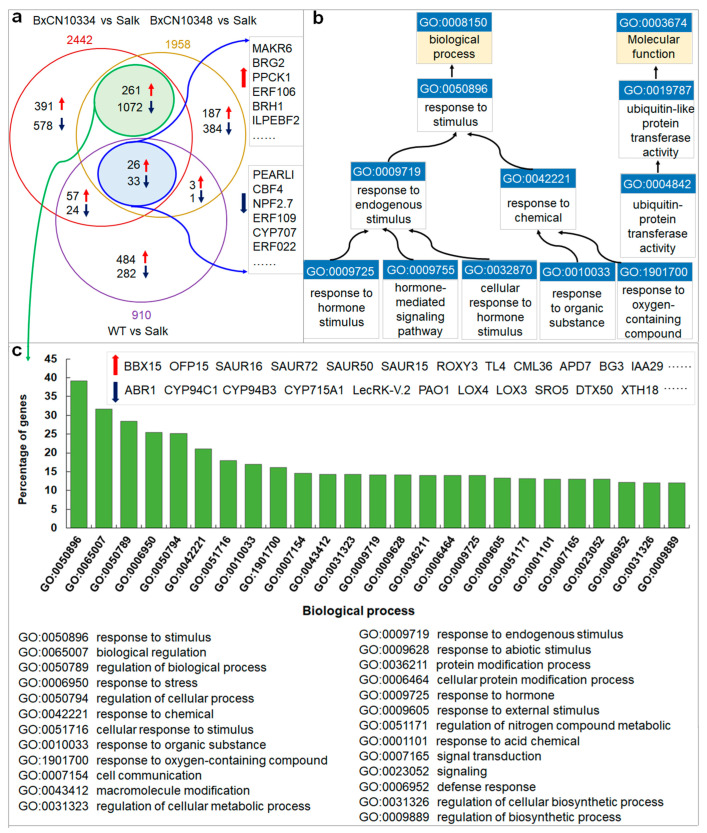
Differentially expressed genes (DEGs) in the four types of *Arabidopsis* transcriptomes and enrichment in GO terms: (**a**) Venn diagram shows common and specific DEGs among the transcriptomes of the wild-type and the two transgenic lines introduced with *B. xylophilus*-specific legumain-coding genes (BxCN10334 and BxCN10337), each compared with the control (*∆γvpe* mutant); (**b**) enriched GO terms of the 59 common DEGs, which with identical expression patterns among the above three transcriptomes; (**c**) the top enriched GO terms of additional 1333 DEGs, which have identical expression patterns between the two transgenic lines transcriptomes.

## Data Availability

The genome sequences of BxCN, BxCA, and BmCN are publicly available in National Genomic Data Center (NGDC) with BioProject accession number of PRJCA011732 (https://ngdc.cncb.ac.cn/bioproject/browse/PRJCA011732). The accession numbers of BxCN, BxCA and BmCN genomes in NGDC are GWHBKKB00000000, GWHBKKC00000000, and GWHBKKD00000000, respectively. Meanwhile, the genome sequences of BxCN are deposited in NCBI, with the accession number POCF00000000 (https://www.ncbi.nlm.nih.gov/nuccore/POCF00000000.1/).

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
