# Peer review of "Roles of Species-Specific Legumains in Pathogenicity of the Pinewood Nematode *Bursaphelenchus xylophilus"

_ijms, 2022, doi:10.3390/ijms231810437_

Round 1
Reviewer 2 Report
In the present study, the authors explored the diversities of peptidases in different plant-parasitic nematodes by comparing peptidases in plant-parasitic nematode genomes. Furthermore, they identified Bursaphelenchu xylophilus-specific legumains and estimated their potentials involved in nematode-plant interaction by experimental verification. A molecular pathogenic mechanism of B. xylophilus killing pine tree was also proposed. The results of this study may provide useful information of legumains in plant-parasitic nematodes. In my opinion, the paper is suitable for publication in its current form after a couple of minor changes:
Authors should specify the assessment of the purity, concentration and integrity of the RNA used. In addition, they should specify the reaction conditions for RT-PCR and quantitative real-time PCR. Authors should also include a couple of lines suggesting directions for future experimental research.
Author Response
Response: Thanks for your comments and suggestion. About the RNA purity, concentration and integrity, we have added related information in “4.6. RNA isolation,RT-PCR and quantitative real-time PCR” section (in page 20 lines 579-603). We also rewrote the discussion section and added suggestion and prospects in future studies (in page 17, lines 449-462; page 18, lines 482-486.

Round 2
Reviewer 1 Report
Most of the points raised by this reviewer has been adequately addressed in the revision. However, the question #3 regarding figure 7 was not resolved but simply its mention was deleted in the discussion section. The result shown in figure 7 still cannot be explained because it was localized basically in the cytoplasm even though it is a secreted protein. Personally I would rather delete the figure 7 entirely if authors cannot offer any reasonable explanation for that because the result will just undermine the entire quality of the work.
Author Response
Most of the points raised by this reviewer has been adequately addressed in the revision. However, the question #3 regarding figure 7 was not resolved but simply its mention was deleted in the discussion section. The result shown in figure 7 still cannot be explained because it was localized basically in the cytoplasm even though it is a secreted protein. Personally I would rather delete the figure 7 entirely if authors cannot offer any reasonable explanation for that because the result will just undermine the entire quality of the work.
Response: Thanks for your comments and suggestions. In the former versions, we suggested these nematode legumains are cytoplasm effectors because no nuclear localization signal (NLS) motif is identified in these legumains. In this version, we respect the reviewer’s view and have deleted all words and the Figure (Fig. 7) related to subcellular localization in host, including those in the sections of results (p.13), discussion (p.16), methods (p20, 4.10), and references (Shi et al. 2018, Merkle et al. 1996). We think perhaps we need more experiments and clear photos to determine the localization of these proteins in host cells.
Reviewer 2 Report
The authors responded satisfactorily to my comments.
Author Response
Thanks for your comments and suggestions.